# Structural characterization of a novel human adeno-associated virus capsid with neurotropic properties

Hung-Lun Hsu [1,2,7], Alexander Brown[1,2,7], Anna B. Loveland [3,7], Anoushka Lotun[1], Meiyu Xu[1,2], Li Luo[1,4], Guangchao Xu[1,4], Jia Li[1], Lingzhi Ren[1], Qin Su[1,5], Dominic J. Gessler[1,2], Yuquan Wei[4], Phillip W. L. Tai [1,2✉], Andrei A. Korostelev [3✉] & Guangping Gao [1,2,6✉]

Recombinant adeno-associated viruses (rAAVs) are currently considered the safest and most reliable gene delivery vehicles for human gene therapy. Three serotype capsids, AAV1, AAV2, and AAV9, have been approved for commercial use in patients, but they may not be suitable for all therapeutic contexts. Here, we describe a novel capsid identified in a human clinical sample by high-throughput, long-read sequencing. The capsid, which we have named AAVv66, shares high sequence similarity with AAV2. We demonstrate that compared to AAV2, AAVv66 exhibits enhanced production yields, virion stability, and CNS transduction. Unique structural properties of AAVv66 visualized by cryo-EM at 2.5-Å resolution, suggest that critical residues at the three-fold protrusion and at the interface of the five-fold axis of symmetry likely contribute to the beneficial characteristics of AAVv66. Our findings underscore the potential of AAVv66 as a gene therapy vector.

[1] Horae Gene Therapy Center, University of Massachusetts Medical School, Worcester, MA, USA. [2] Department of Microbiology and Physiological Systems, University of Massachusetts Medical School, Worcester, Massachusetts, USA. [3] RNA Therapeutics Institute, University of Massachusetts Medical School, Worcester, MA, USA. [4] State Key Laboratory of Biotherapy, West China Hospital, Sichuan University, Chengdu, P. R., China. [5] Viral Vector Core, University of Massachusetts Medical School, Worcester, MA 01605, USA. [6] Li Weibo Institute for Rare Diseases Research, University of Massachusetts Medical School, Worcester, MA, USA. [7] These authors contributed equally: Hung-Lun Hsu, Alexander Brown, Anna B. Loveland. ✉email: phillip.tai2@umassmed.edu; Andrei.Korostelev@umassmed.edu; guangping.gao@umassmed.edu

Adeno-associated viruses (AAVs) are non-enveloped ~25 nm viruses that encapsidate a 4.7-kb single-stranded DNA (ssDNA) genome[1]. AAVs belong to the dependo-parvovirus genus of the parvoviridae family of small DNA viruses. The wild-type AAV genome encodes four open reading frames *rep*, *cap*, assembly activating protein (*AAP*), and the recently discovered membrane-associated accessory protein (*MAAP*)[2,3]. The AAV *cap* gene encodes for three capsid subunits, VP1–VP3 that assemble into a $T = 1$ icosahedral capsid at an approximate ratio of 1:1:10, respectively[2]. VP1 contains a unique N-terminal domain (VP1u) that harbors a phospholipase A2 domain, which is believed to be responsible for endosomal escape[4]. VP1u and VP2 domains contain nuclear localization signals that are also required for viral infection[5,6]. VP3, the major subunit of the capsid, establishes tissue tropism via cell receptor recognition and antigenic response by the host[7–11]. Diversity of serotypes has been generally defined by amino acid differences within nine variable regions found in VP3 (VR-I to VR-IX)[12].

AAVs have recently attracted attention as effective and proven gene therapy vectors. The current class of AAV vectors confers stable, long-term gene expression, have a broad range of tissue tropisms, and exhibit relatively low pathogenicity[2,13–15]. To date, three serotype capsids (AAV1, AAV2, and AAV9) have gained regulatory approval for commercial use in patients[2]. Unfortunately, the current library of discovered and engineered AAV capsids falls short for certain clinical applications that require targeting of specific tissues or cell types[2]. Furthermore, patients may have pre-existing immunity to the vector via neutralizing antibodies (NAbs) that would limit therapeutic efficacy. In addition, certain capsids are known to be problematic under standard production schemes for generating high-yield titers needed to meet therapeutic doses. In response to these shortcomings, there is a need to discover and develop capsids that exhibit better vector yields, can escape innate immunity, and possess unique tropism profiles.

Various strategies have been utilized to develop novel capsids. These methods can be divided into four broad approaches: natural discovery, rational design, directed evolution, and in silico discovery[16–21]. Natural discovery in nonhuman primates and humans has led to the identification of vectored AAVs that have shown success in preclinical and clinical studies[2]. In this report, we describe a novel capsid that was identified by high-throughput single molecule, real-time (SMRT) sequencing[22] and whose properties substantially differ from those of AAV2, despite high (98%) sequence similarity. First, AAVv66 exhibits a better vector yield and is more thermostable than prototypical AAV2. Second, AAVv66 has a better distribution spread within brain tissue when administered by intracranial and systemic injections. Finally, AAVv66 shows some antigenic distinction from AAV2. To better understand how AAVv66 differs from AAV2, we performed cryogenic electron microscopy (cryo-EM) to explore the structural and functional characteristics that define AAVv66. Our 2.5-Å resolution structure of the AAVv66 capsid reveals differences from the structure of AAV2 and provides insights into the functional properties the capsids. Together, our observations elucidate the mechanistic properties of AAVv66 and enhance the understanding of biophysical divergence across natural AAV capsids.

## Results

**Identifying AAV variants in human tissue by SMRT sequencing.** To identify full-length capsid sequences from human tissues, we performed SMRT sequencing to obtain long DNA reads that span the entire capsid open reading frame (Fig. 1a). This method can resolve sequences of long DNA fragments without the need for sequence assembly, which is necessary for short-read sequencing approaches. In this way, capsid diversity, which is governed by point mutations and recombination events, can be assessed on individual intact molecules that span the entire capsid open reading frame (ORF). To explore AAV diversity, we selected a single tissue from about 800 human surgical samples that were collected previously[23]. Using primers that flank the capsid ORF at conserved sequence across known serotypes, we generated target polymerase chain reaction (PCR) amplicons for SMRT sequencing analysis. From the single tissue, a total of 16,681 ~2.2-kb reads were obtained. From this set, we identified 361 full-length and unique AAV proviral capsid sequences, which translated to 81 unique amino acid sequences. Upon tabulating unique amino acid sequences, we found one capsid sequence that made up ~45% of all sequences identified from the single tissue (Fig. 1b). This predominant capsid, tentatively named "variant 66" (AAVv66), exhibits closest homology to AAV2 (98% sequence similarity; Fig. 1c, d). We found that AAVv66 contains 13 amino acid residues that differ from AAV2 (Fig. 1c, Supplementary Fig. 1a): one within the VP1u region (K39Q), one within the VP2 domain (V151A), and eleven within VP3 (R447K, T450A, Q457M, S492A, E499D, F533Y, G546D, E548G, R585S, R588T, and A593T). Notably, the unique amino acid residues within VP3 are all in or near variable regions VR-IV through VR-VIII.

To better make predictions for the contributions of the unique AAVv66 amino acid residues toward capsid function, the VP3 region was compared with those of other contemporary AAV serotypes (AAV1–AAV9). The most notable differences occur at four positions (499, 533, 585, and 588), which are highly conserved among AAV serotypes (Supplementary Fig. 1b). At position 499, most serotypes harbor an asparagine, while AAVv66, AAV2, AAV4, and AAV9 have a negatively charged aspartic acid or glutamic acid. The highly conserved phenylalanine at position 533 is a tyrosine in AAVv66 (also, T533 in AAV5). Finally, unlike AAV2, which harbors positively charged arginine residues that define AAV2's capacity to bind heparan sulfate proteoglycans (HSPG) at positions 585 and 588[11], AAVv66 contains S585 and T588 (identical to AAV1, AAV3, AAV5, and AAV6).

**AAVv66 and AAV2 differ in production and cell infectivity.** The strong affinity of AAV2 for heparin and its resulting strong cell–surface association is proposed to lead to the virus' relatively poor packaging titers[24]. The limited vector yield by AAV2 is thought to result from non-productive binding and re-infection of the packaging cells by vector particles during production. We therefore sought to compare vector production and cell infectivity of AAVv66 with those of AAV2 and AAV3b. Of note, AAV3b is the closest distinct cousin of AAV2 (89% sequence similarity), but uses different electrostatic surface charges at the threefold protrusions to weakly bind heparin[25]. This difference between AAV3b and AAV2 likely explains AAV3b's increased packaging titers that result from weaker transduction of HEK producer cells during production[26,27].

We first compared the packaging profiles of AAVv66 with those of AAV2 and AAV3b by measuring the yields of encapsidated vector genomes in cell lysates. To this end, the AAVv66 capsid ORF was synthesized and cloned into a *trans* plasmid expressing AAV2 *rep* under the AAV2 *p5* promoter (pAAV2/v66). Small-scale vector preparations of AAVv66, AAV2, and AAV3b were used to package a single-stranded vector consisting of the firefly luciferase transgene driven by the ubiquitous chicken-beta actin promoter (AAV-*CB6-Fluc*)[28]. Quantification of viral vector yields by crude lysate qPCR[29]

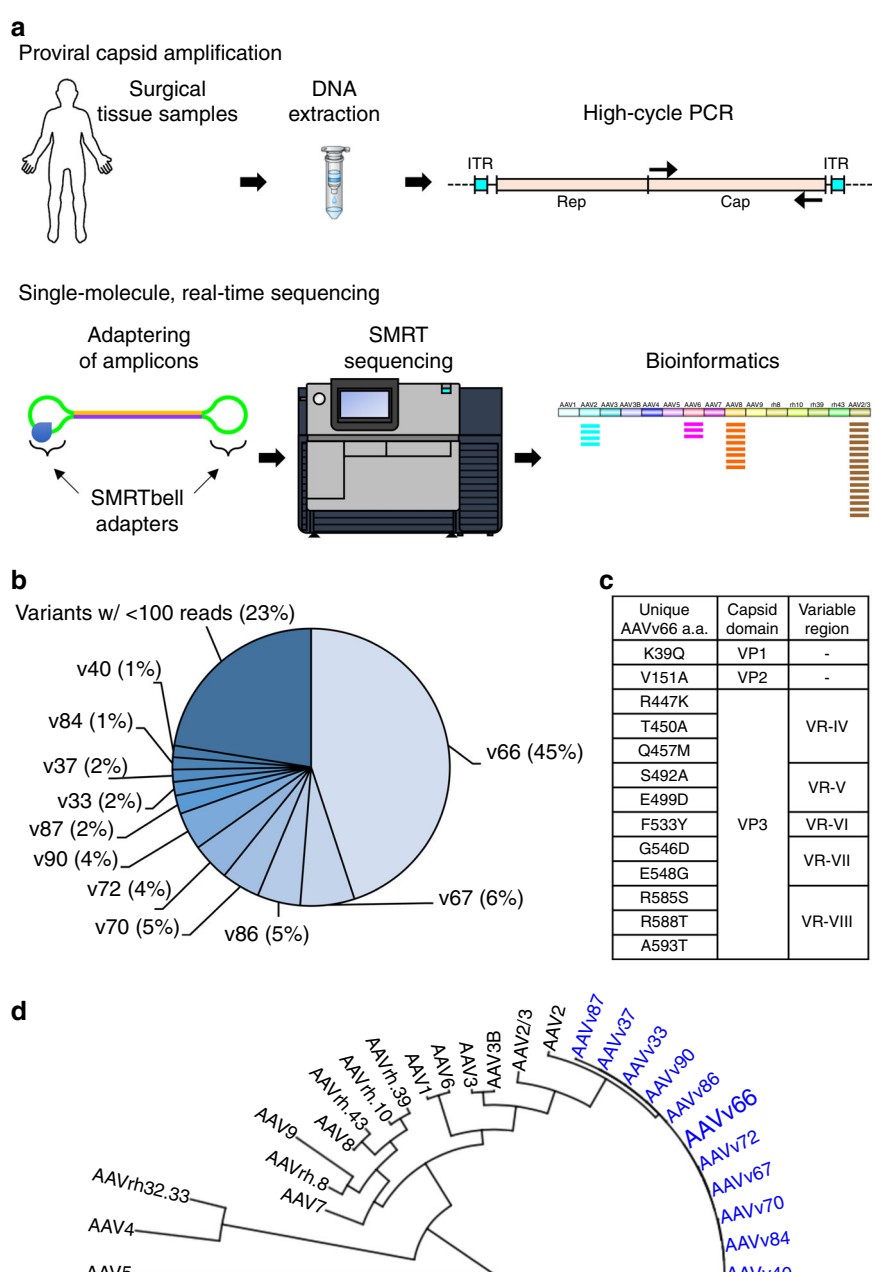

**Fig. 1 Identification of a novel proviral AAV capsid sequences from a human surgical sample. a** AAV capsid proviral sequences were first PCR amplified from a human surgical sample using primers that flank the AAV *cap* ORF. Amplicons were subjected to single molecule, real-time (SMRT) sequencing and the resulting reads were analyzed by BWA-MEM alignment to contemporary AAV serotype sequences, InDelFixer to remove insertion/deletions related to PCR or SMRT sequencing errors, and de novo assembly to cluster reads of high sequence similarity. **b** The *cap* sequence of variant AAVv66 was found to be the most abundant in the analysis (45%). **c** Summary of the 13 unique residues in the AAVv66 capsid sequence that are different from AAV2. **d** Phylogenetic tree of AAV2 variants reported in this study (blue) and contemporary serotypes.

revealed that the production of encapsidated, DNase-resistant AAVv66 vector genomes is ~2.4-fold higher than AAV2 yields and is ~30% higher than AAV3b yields (Supplementary Fig. 2). We next asked whether the higher abundance of AAVv66 in crude lysate is due to nonproductive binding of particles to the packaging cells and/or reinfection of the cells, which would be revealed by a predominance of AAVv66 particles in the media rather than in the cell lysate fraction. PCR analysis showed that encapsidated genomes of AAVv66 within the media are approximately threefold more abundant than in cell lysates (Supplementary Fig. 2). In contrast, very few AAV2 particles were detected in the media of packaging cells, consistent with previous

reports[24]. To test whether AAVv66's ability to produce more DNase-resistant genomes is related to poor HSPG binding and weak re-infectivity of packaging cells, a heparin competition assay was performed (Supplementary Fig. 3). For this purpose, large-scale AAVv66 and AAV2 vectors, again packaging *CB6-Fluc*, were produced using the standard cesium chloride purification protocol[30]. We found that the transduction of AAVv66 is not affected by the presence of heparin, whereas 1.25 µg per well of heparin blocked AAV2 transduction by 50% and 5 µg per well of heparin completely abolished transduction. These results suggest that the improved production efficiency of AAVv66 is, at least in part, due to poor heparin binding.

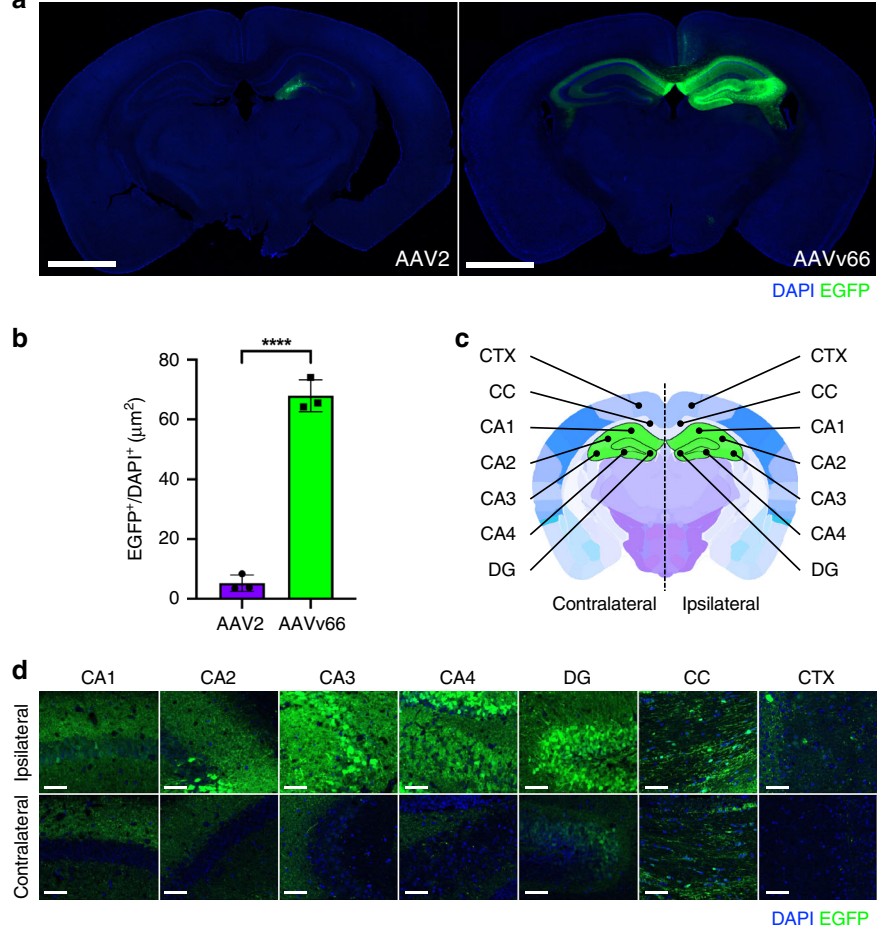

**Fig. 2 Transduction spread of rAAV2 and rAAVv66 following intrahippocampal injection. a** Representative native EGFP (green) expression following rAAV2-CB6-PI-*EGFP* or rAAVv66-CB6-PI-*EGFP* injection via unilateral intrahippocampal administration. Scale bars = 700 μm. **b** Quantification of EGFP-positive surface normalized to DAPI-positive surface from panel **a** data. Data are presented as the mean ± SD; $n = 3$ animals/group from one experiment. ****$P < 0.0001$, two-tailed Student's $t$ test. **c** Coronal brain schematic depicting sub-anatomical regions of interest in both contralateral and ipsilateral hemispheres (adapted from Allen Mouse Brain Atlas). Cornu ammonis (CA1–CA4), dentate gyrus (DG), corpus callosum (CC), and cortex (CTX). **d** Representative high-magnification images of rAAVv66 transduced sub-anatomical regions from $n = 3$ animals from one experiment. Scale bars = 50 μm.

To determine whether AAVv66's lower affinity to heparin coincides with reduced cell transduction compared to AAV2, purified AAVv66, AAV2, and AAV3b vectors were used to infect HEK293 cells. As expected, AAV2 exhibits greater transduction than AAVv66 (~65-fold) and AAV3b (~7.5-fold) (Supplementary Fig. 4). Thus, our results indicate that the vectorized AAVv66 proviral capsid is able to efficiently transduce cells in vitro, but its vector production and cell infectivity properties are distinct from those of its closest serotype relative, AAV2.

**AAVv66 exhibits central nervous system (CNS) transduction that is distinct from AAV2.** We next asked whether the AAVv66 capsid has potential utility as a clinical vector. We tested AAVv66 for its capacity to transduce selected target tissues via different routes of administration. To this end, we assessed the biodistribution of AAVv66 in mice via multiple routes of delivery (Supplementary Figs. 5 and 6). Among all routes tested, the most striking was AAVv66's transduction profile following intracranial delivery to target cells of the central nervous system (CNS) (Fig. 2). To determine whether AAVv66 has increased tropism in the CNS, relative to that of AAV2, we packaged into the AAVv66 and AAV2 capsids the *EGFP* transgene driven by the CB6 promoter[28]. Vectors were unilaterally injected into the right hemisphere of the hippocampus at a dose of 3.6E9 vector genomes (vg)

per animal. Four weeks post-injection, cryo-sections of treated brains showed that AAVv66 transduced ~13-fold more cells of the CNS than AAV2, as demonstrated by the enhanced spread throughout the tissue, while AAV2 tended to stay localized to the site of injection (Fig. 2a, b). High-magnification imaging of contralateral regions to the site of injection showed that all sub-anatomical regions of the brain (cornu ammonis [CA1–CA4], dentate gyrus, and corpus callosum, Fig. 2c), exhibited detectable levels of EGFP expression (Fig. 2d), indicating that AAVv66 can spread efficiently throughout the hippocampal hemispheres.

We next aimed to determine the specific cell types that were transduced by AAVv66. Antibody staining was performed with cell type-specific markers; anti-NEUN (neurons), anti-GFAP (astrocytes), anti-IBA1 (microglia), and anti-OLIG2 (oligodendrocytes) (Fig. 3a, e, i, m). 3D-volume reconstruction of sub-anatomical CNS regions demonstrates that EGFP expression colocalized with each investigated cell type (Fig. 3b, f, j, n). As expected, neurons were the predominant cell type found in the cortex and CA1 regions (Fig. 3c). Interestingly, CA2–4 regions and the dentate gyrus exhibited the greatest transduction (~20–40%). Astrocytes and microglia shared a similar distribution pattern, showing the highest enrichment in the dentate gyrus (Fig. 3g, k). Astrocytes showed approximately 1–7% transduction across all regions (Fig. 3h), while microglia exhibited slightly

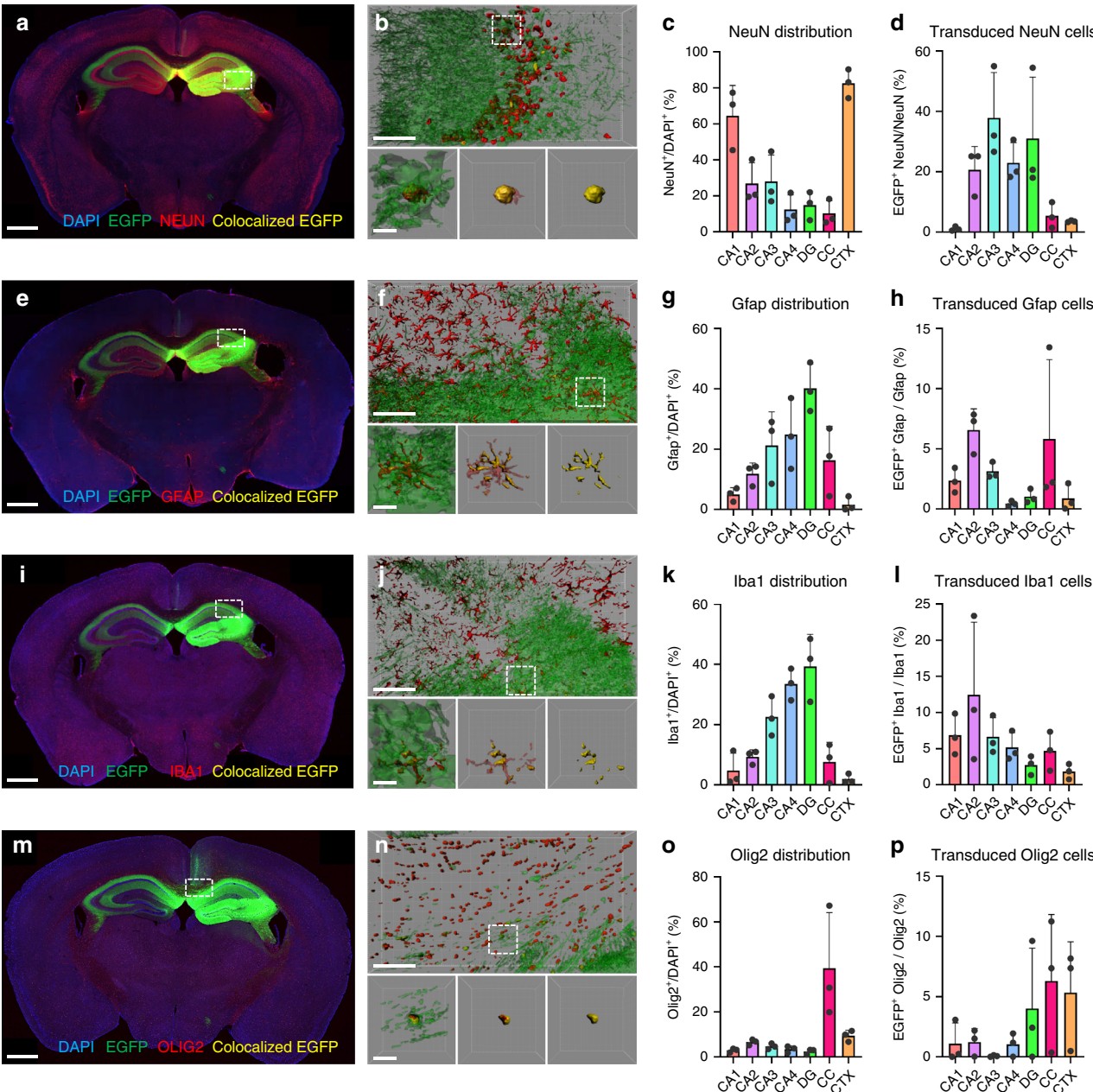

**Fig. 3 Transduction of major cell types of the brain by rAAVv66. a, e, i, m** Coronal sections of rAAVv66-*CB6-PI-EGFP* transduced mouse brains. IF-stained sections (red) with antibodies against NEUN (**a,** neurons), GFAP (**e,** astrocytes), IBA1 (**i,** microglia), or OLIG2 (**m,** oligodendrocytes) indicate the distribution of cell types across the brain. Native EGFP expression (green) that colocalize with IF staining (yellow) reveal the positively transduced cell type indicated. Scale bars = 700 μm. **b, f, j, n** 3D rendering of sub-anatomical regions of single representative frames from dashed line rectangle boxes within coronal section views (top panels) with single-cell representations from fields defined by dashed lined square boxes (bottom three panels). Left panels, total area EGFP and cell marker IF stains; center panels, colocalized EGFP with total cell marker IF stains; right panels, colocalized EGFP and cell marker IF stains. Scale bars = 50 μm (top panels), 5 μm (bottom three panels). **c, g, k, o** Quantification of cell type-specific IF staining across indicated hippocampal regions (x-axes), normalized to DAPI signal. **d, h, l, p** Quantification of cell type-specific transduction across indicated regions, normalized to total cell-type IF and DAPI signal. Data are presented as the mean ± SD, n = 3 animals/group from one experiment. Cornu ammonis (CA1–CA4), dentate gyrus (DG), corpus callosum (CC), and cortex (CTX).

higher transduction efficiencies (2–12%) (Fig. 3l). Oligodendrocytes were enriched in the corpus callosum (Fig. 3o) and had approximately 1–7% transduction by AAVv66 across all regions (Fig. 3p). These findings reveal that AAVv66 can transduce all major cell types of the mouse CNS following intrahippocampal injection.

One desirable characteristic of capsids that can transduce the CNS, like AAV9, is their ability to traverse the blood brain barrier

(BBB)[31]. To determine whether AAVv66 also has this capacity, we injected AAVv66-*EGFP* vector into neonatal mice (4E11 vg per mouse) by facial–vein administration. AAV2, which poorly traverses the BBB and AAV9, the gold-standard serotype for bypassing the BBB, were also tested. Mice were sacrificed after one month and brains were isolated, cryo-sectioned, and stained with anti-GFP antibody to reveal the extent that AAVv66 can transduce the CNS following intravenous vector delivery

(Supplementary Fig. 7). Transduction of AAVv66-*EGFP* (Supplementary Fig. 7e) lead to capsid distribution approaching half of what was conferred by AAV9 (Supplementary Fig. 7i). AAVv66 transduction substantially outperformed that of AAV2. These findings suggest that AAVv66's potency as a neurotropic vector is best achieved via intracranial delivery.

**AAVv66 is serologically distinct from AAV2.** Neutralization of AAV by the host immune system is a major limiting factor for AAV vector transduction efficacy. Individuals harboring pre-existing antibodies against AAV serotypes used as capsids in therapeutic vectors are at a greater risk to adverse effects and ineffectual treatment. Furthermore, patients requiring repeated administration of an AAV gene therapy, risk poorer transduction efficiencies, and stronger immune responses upon read-ministration, necessitating alternative vectors. We therefore sought to examine whether AAVv66 transduction can be blocked by pre-immunization with AAV2. To create pre-existing anti-AAV2 antibodies in circulation, AAV2-*EGFP* vectors (1E11 vg per mouse) were intramuscularly delivered to mice. Sera were collected after 4 weeks to assess NAb titers in vitro (Supplementary Fig. 8a, b). As expected, low NAb titers were needed to achieve 50% neutralization ($NAb_{50}$) of AAV2 infection in Huh-7.5 cells (1/1280–1/2560). This outcome demonstrates that antibodies generated from AAV2 pre-immunization are sufficient to inhibit AAV2 transduction in vitro. By contrast, the $NAb_{50}$ for AAVv66 infection with AAV2-treated mouse sera was 1/20–1/40, indicating that AAVv66 is able to infect cells despite the presence of NAbs generated against AAV2. To test these findings with a secreted therapeutic transgene product in vivo, we re-dosed AAV2-immunized mice with AAV2 or AAVv66 packaged with the alpha-1 antitrypsin transgene (AAV2-*A1AT* or AAVv66-*A1AT*). Sera were collected at weeks 5–8, and secreted A1AT levels were quantified by enzyme-linked immunosorbent assay (ELISA)[32]. Low A1AT expression would suggest that NAbs generated from the first vector dose prevents the transduction of the second vector dose. To establish a baseline for maximal A1AT expression, naïve mice were also treated in the same fashion. At weeks 6 and 7, A1AT expression in mice treated with AAV2-*EGFP* and then AAVv66-*A1AT* reached ~90% of A1AT expression as compared to naïve mice, whereas mice redosed with AAV2-*A1AT* reached only ~40% of naïve levels (Supplementary Fig. 8c). These results concur with our in vitro observations of robust infectivity by AAVv66 in the presence of sera from mice pre-immunized with AAV2 capsids. However, when the in vitro NAb assay was repeated using sera from mice pre-immunized with AAVv66, transduction by both AAV2 and AAVv66 was substantially inhibited (Supplementary Fig. 9a, b). Notably, the AAVv66-treated mouse sera is more potent in inhibiting AAVv66 ($NAb_{50}$ ~1/2560) than AAV2 ($NAb_{50}$ ~1/1280). This observation suggests that the serology of AAVv66 is not entirely distinct from AAV2.

In the following series of experiments, we tested whether pre-immunity by a broad range of AAV serotypes (AAV1, AAV2, AAV3b, AAV8, AAV9, AAV-DJ, AAVrh.8, and AAVrh.10) can limit AAVv66 vector transduction. Antisera of rabbits separately pre-immunized with the eight serotypes were screened for AAVv66-vector neutralization. We found that, AAV1, AAV3b, and AAV-DJ exhibit about an order of magnitude difference in $NAb_{50}$ titers compared to AAVv66, whereas AAV2, AAV8, AAVrh.8, and AAVrh.10 exhibit a two-order magnitude difference, and AAV9 had a three-order magnitude difference (Supplementary Fig. 8d). Taken together, these findings demonstrate that AAVv66 has some serological overlap with AAV2, but is distinct from other contemporary AAV capsids.

**The AAVv66 capsid is more thermostable than AAV2.** The efficient formation and structural stability of the capsid is essential to the production, purification, and storage of viral vectors. In addition, for productive infection to take place, vector particles must maintain stability throughout the entry process and must uncoat only under conditions in which delivery of the genomic payload can result in transduction of the cell. Although AAV vectors have been studied widely and are utilized for their strong transduction profiles in a range of tissues, the processes of intracellular trafficking, endosomal escape, and transportation of capsids into the nucleus are not fully understood. Among the presumed intracellular checkpoints impacting AAV intracellular trafficking and transduction that are dependent on capsid dynamics, endosomal escape is best understood. This process is believed to be triggered by a pH-dependent structural change of the capsid. Acidification of the endosomal lumen leads to a conformational change of the VP1 domain and exposure of the PLA2 domain within VP1, which triggers escape from the endosome compartment[33]. In principle, vector capsids that can retain stability throughout intracellular trafficking are desirable and are expected to have high transduction capacities.

To determine the overall stability of AAVv66 capsids, differential scanning fluorimetry (DSF) analysis was used to measure the thermostability of the AAVv66 capsid across a range of physiological pHs (pH7–pH4) (Fig. 4)[34,35]. This range includes pH 4.5, which is observed in the lumen of late endosomes and lysosomes[36]. In this assay, vector particles are suspended in SYPRO Orange dye, which fluoresces upon binding to hydrophobic residues in proteins. Thus, peak fluorescence signals are an indirect readout for maximally bound hydrophobic regions exposed upon protein unfolding. The melting temperatures (maximum slope values [$\Delta$signal/$\Delta$temp], $T_m$) for AAVv66 are more than five degrees higher than for AAV2 across all pH conditions tested. The most extreme difference was observed at pH 7, where the $T_m$ of AAVv66 (75.29 ± 0.34 °C) is nearly 10° higher than that of AAV2 (65.85 ± 0.18 °C) (Fig. 4a). Thus, the AAVv66 capsid is more thermally stable and resistant to pH than AAV2.

We next measured how the stability of AAVv66 capsid affects vector genome release. Gauging vector genome release as a function of temperature range has been used as a proxy for pressure-driven DNA extrusion exerted by the nucleolar environment[34]. Although some parvoviruses have been reported to release genomes within endosomes as a possible prerequisite to nuclear entry[37], whether capsid uncoating occurs for AAV virions under the acidic environment of endosomal lumen is relatively understudied. Therefore, we compared the temperature dependence of AAVv66 genome release with that of AAV2 under different pHs. To this end, we employed DSF analysis with SYBR Gold dye, which fluoresces upon binding to DNA. Peak fluorescence is an indirect measure of maximal accessibility of the encapsidated genomes to the dye solution. Vector genome release at pH 7 was observed to be concomitant with capsid stability, showing signal peaks at ~65 °C for AAV2 and ~74 °C for AAVv66. At lower pHs, however, peak fluorescence for dye-accessible DNA was detected at lower temperatures than peak fluorescence for unfolded capsid protein (Fig. 4b). Furthermore, DNA accessibility for AAVv66 was more evident than that for AAV2, where peak DNA accessibility at pHs 5 and 4 for AAV2 occurred at ~53 °C and ~42 °C, respectively; and AAVv66 exhibited peak signals at 25 °C. This surprising observation shows that genomes packaged into AAVv66 capsids are especially accessible at low pHs (4–5), even at room temperature.

We next asked how AAVv66-specific amino acid residues contribute to the structural and functional differences observed

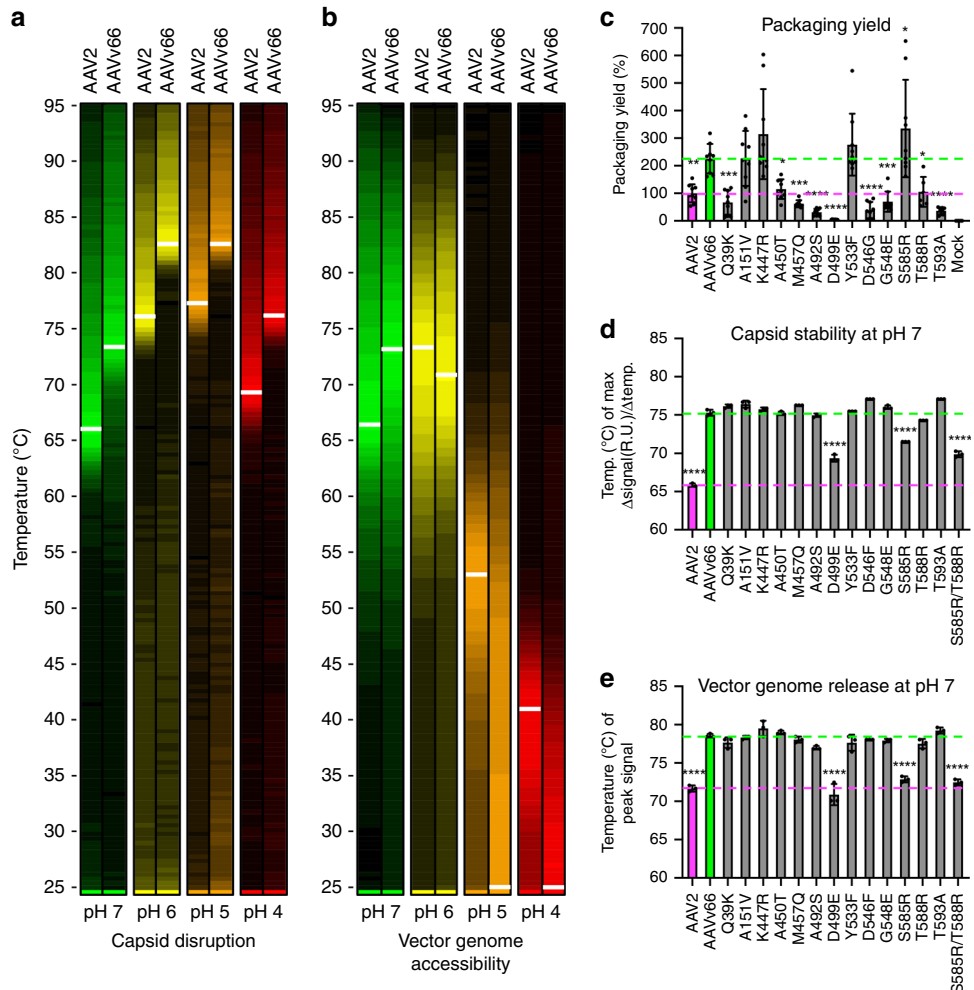

**Fig. 4 Biophysical analyses of AAVv66. a, b** Heatmap displays of differential scanning fluorimetry (DSF) analyses to query (**a**) capsid protein unfolding (uncoating) and (**b**) DNA accessibility (vector genome extrusion) at pHs 7, 6, 5, and 4. Color scaling depicted represent relative peak signals from highest to lowest value (brightest to dimmest, respectively). **c–e** Each defining amino acid residue of AAVv66 was converted to those of AAV2 by site-directed mutagenesis and examined for changes in **c** packaging yield, **d** capsid stability, and **e** genome release at pH 7. Values represent mean ± SD. $p$ Values were determined by one-way ANOVA. *$p < 0.05$, **$p < 0.01$, ***$p < 0.001$, ****$p < 0.0001$. $n = 3$/group.

between AAVv66 and AAV2. We individually mutated the thirteen AAVv66-defining amino acid residues to those of AAV2 and tested their impact on packaging vector genomes during production with HEK293 cells (Fig. 4c). In addition, thermal capsid stability and vector genome release were assessed for the mutant capsids (Fig. 4d, e). All but four mutations (A151V, K447R, Y533F, and S585R) resulted in yields of DNase-resistant genomes that were similar to or lower than those of AAV2 (Fig. 4c). Remarkably, the relatively conservative mutation D499E, which does not result in a change in charge, lowered the packaging yield to ~5% of AAV2 yields. The modification also affected capsid stability, as D499E along with S585R and the S585R/T588R double mutation lowered the $T_m$ by 5.9, 3.8, and 5.4 °C, respectively (Fig. 4d), while other mutations affected the $T_m$ by only 1–2 °C. Vector genome accessibilities of these three amino acid mutations exhibited lowered peak signal temperatures, while other mutations led to little or no change (Fig. 4e). Notably, the overall titers of purified vector were not drastically impacted (Supplementary Table 1). Thus, the packaging yield of AAVv66 is only partially dependent on capsid stability, suggesting that partial capsid destabilization may be sufficient to facilitate genome release. Only residue D499 drastically affects both packaging and capsid stability.

**Cryo-EM reveals differences between AAVv66 and AAV2.** To characterize the structural properties of AAVv66, we subjected purified AAV2v66-*EGFP* vector to cryo-EM analysis. We obtained 52,874 particle images, which yielded a cryo-EM map at 2.5 Å resolution (Fig. 5, Supplementary Figs. 10 and 11), and obtained a structural model with optimal real-space fit and stereochemical parameters (Supplementary Table 2). Overall, the AAVv66 structure is similar to AAV2 (root-mean-square deviation of atomic coordinates = 0.456 Å) (Supplementary Fig. 12)[38]. Thus, AAVv66 exhibits the characteristic features of an AAV capsid, which include the depression at the twofold axis, the threefold symmetry that is defined by the threefold protrusions, and the fivefold pore that is comprised of five monomers that form the interface and pore for Rep binding (Fig. 5a). Of note, VP1u and VP2 domains are each represented at approximately a twelvth of the VP3 domains for each particle, and similar to other AAV structures before[7,8,38–41], were not resolved in our symmetrized cryo-EM map. Therefore, only residues 219–736 are definitively resolved within the cryo-EM map, including 11 of 13 AAVv66-defining residues (Fig. 5b).

Comparison of the AAVv66 structure with that of AAV2 reveals several distinctive structural features that may contribute to the differences in DNA packaging and/or capsid stability. Key

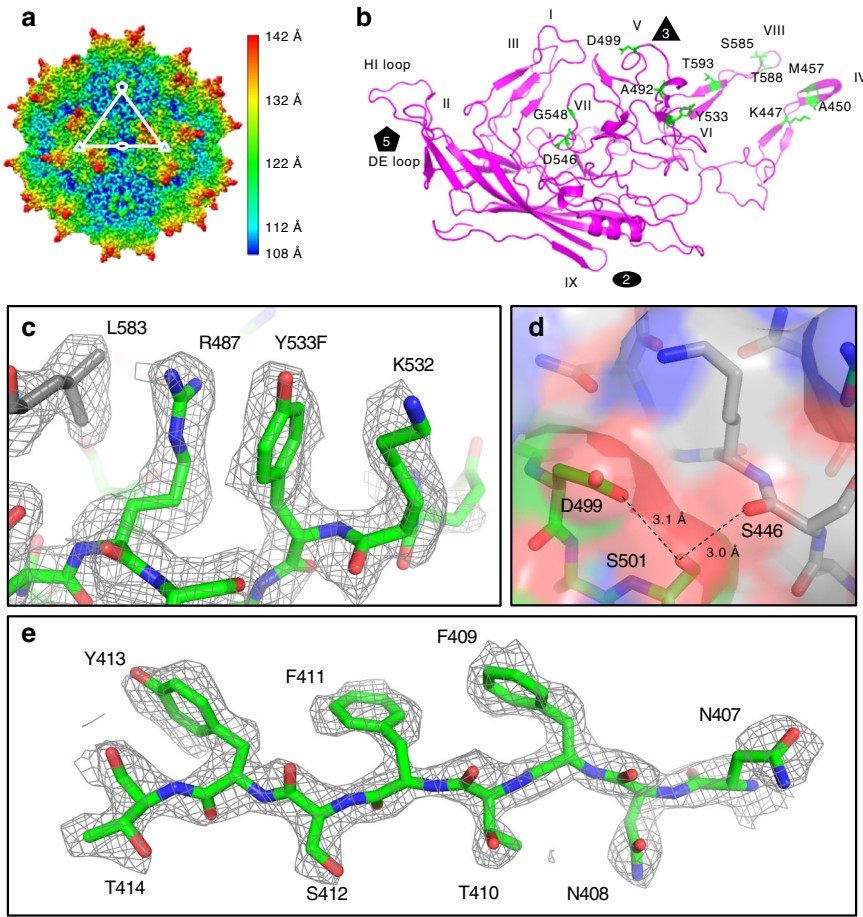

**Fig. 5 Cryo-EM primary metrics, map reconstruction, and model generation of AAVv66. a** Density map of AAVv66. Color scheme demarcates the topological distance from the center (Å). **b** Ribbon structure of the refined AAVv66 capsid monomer. Amino acids differing from AAV2 are highlighted in green. The twofold (oval), threefold (triangle), and fivefold (pentagon) symmetries are annotated. Part of AAVv66 electron density (dark grey mesh) and residues (green for one monomer and grey for neighboring monomers) are shown for regions close to **c** L583, R487, Y533, and K532, **d** S446, D499, and S501, and **e** N407-T414.

differences occur at the interfaces between monomers of VP3 at the protrusions around the threefold axis. D499, whose mutation to the longer glutamate residue resulted in dramatic defects in vector genome packaging (Fig. 4c), forms electrostatic interactions and/or hydrogen bonding with S501 (Fig. 5d). This region is tightly packed against the neighboring VP3 monomer (Fig. 5d). Here, the backbone atoms of D499 and S501 respectively interact with the side chains of the symmetry-related N449 and T448 amino acids (Fig. 6c, Supplementary Table 3), whereas the hydroxyl group of S501 hydrogen-bonds with the backbone carbonyl of the symmetry-related S446. Our observation of the strong effect of D499 mutation is therefore likely due to the disruption of the interface between VP3 monomers, leading to destabilization of the capsid. In the same region, residues K447 and A450 of the neighboring monomer eliminate potential for electrostatic interaction between corresponding AAV2-R447 and T450 side chains (Fig. 6d). Amino acid M457 is located on the three-fold protrusion of AAVv66 at variable region IV, with the sidechain poised toward the solvent (Fig. 6e). Interestingly, this methionine is a unique feature among other serotypes (Supplemental Fig. 1b), suggesting at potential unique capsid interactions with cellular receptors, host factors, or antibodies. The polar hydroxyl group in AAVv66-Y533 (AAV2-F533) likely stabilizes the polar environment between the side chains of R487 and K532, and may contribute to the interaction with L583 of the symmetry-related monomer (Fig. 6a). AAVv66-D546 and G548

redistribute the surface charge conferred by AAV2-G546 and E548 (Fig. 6b) and is yet another defining feature of AAVv66.

A key functional region of AAV2 involves the positively charged arginine residues at position 585 and 588. These residues are at the surface of the threefold protrusions and govern the capsid's interaction with HSPG receptors, which are vital to attachment and entry in many cell types[42]. By contrast, S585 and T588 in the AAVv66 capsid are neutral charged polar residues (Fig. 6f), similar to S586 and T589 of AAV3b (Supplementary Fig. 1b). AAV3b's physical and functional interactions with HSPG rely on electrostatic interactions conferred by residues R447 and R594 (R447 and A593 in AAV2)[8], but AAVv66 also lacks these arginine residues (K447 and T593). These differences from AAV2 and AAV3b suggest that AAVv66 associates differently with the canonical cell surface receptor commonly utilized by AAV clade B and C capsids, consistent with our findings that AAVv66 lacks heparin binding.

**AAV2 and AAv66 show significant surface charge differences.** Because electrostatic properties of the virus are important for capsid-receptor interaction[8,43], we next determined how the net loss of positive charge for the AAVv66 capsid in relation to AAV2 affects the electrostatic properties of the capsid. First, we compared the calculated electrostatic potential values for AAV2 and AAVv66 structures (Fig. 7a). The distribution of electrostatic potential on the surface of AAVv66 differs from that of AAV2.

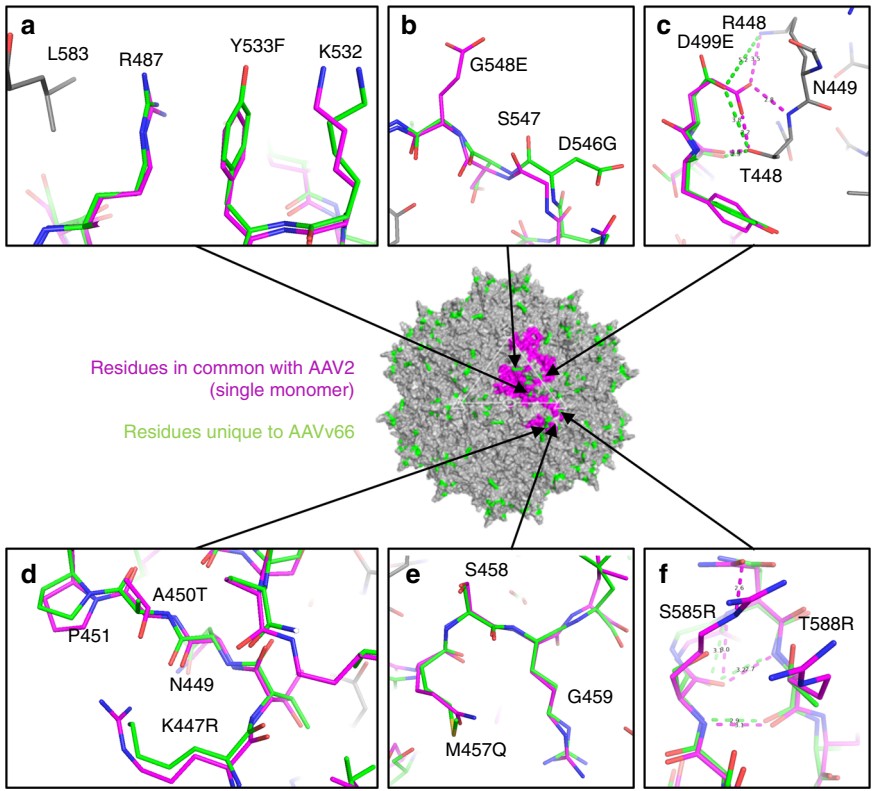

**Fig. 6 Structural differences between AAVv66 and AAV2.** At the center is the AAVv66 60-mer structure (grey). Amino acid residues unique to AAVv66 are highlighted in green, while amino acid residues for a single monomer that are in common with AAV2 are colored in magenta. **a–e** Atomic models showing residue side chains of select regions with substantial difference between AAVv66 and AAV2. The alignments were made using monomers of AAV2 (1lp3) and AAVv66, with modeled side chains from neighboring residues displayed in grey. Annotations for amino acids shown are indicated as those belonging to AAVv66, the position number, and then AAV2. **c, f** Distances between atoms of interest are indicated by dashed lines (AAV2, magenta; AAVv66, green) and reported in angstroms (Å). Distance measurements were conducted in PyMOL and are also reported in Supplementary Table 3.

The most notable difference is at the threefold protrusions, where the positive charge conferred by R585 and R588 in AAV2 is drastically reduced by S585 and T588 in AAVv66 (Fig. 7b).

We next asked whether the distinct structure and surface electrostatics of AAVv66 affects the charge-dependent particle migration (zeta potential) of the capsid (Fig. 7c). The zeta potential of AAVv66 (−10 mV) is remarkably different from that of AAV2 (−3.5 mV), consistent with differences in electrostatic potential between the capsids. To test the contributions of individual substitutions, we have also measured particle migration of AAVv66 harboring single amino acid substitutions that convert residues to the corresponding residues of AAV2 (Fig. 7c). The single mutations S585R and T588R resulted in the most dramatic change of the zeta potential (by ~3 mV each), bringing the zeta potential closer to that of AAV2 (Fig. 7c). These observations indicate that the electrostatic properties of AAVv66 are distinct from those of AAV2 and the difference is predominantly due to substitutions at positions 585 and 588. Thus, interactions of capsid AAVv66 with receptors, antibodies, and other proteins likely differ substantially from those of other closely related capsids.

## Discussion

Capsid variants identified from natural isolates are valuable as therapeutic candidates, because their persistence within the natural population implies low immunogenicity and high structural and functional integrity. Here, we used long-read SMRT sequencing to obtain full-length proviral genomes from a single tissue sample. Among the highly abundant AAV isolates, we observed significant diversity at the DNA and amino acid primary sequences. Importantly, we have identified a novel variant named AAVv66 that boasts strong packaging performance and is functionally distinct from AAV2, despite sharing 98% sequence similarity. As a naturally occurring variant, AAVv66 may become a promising gene-therapy vector. Furthermore, our findings provide structural and functional understanding of AAV capsids, which may be informative for future capsid engineering efforts to improve the characteristics of AAV vectors for gene therapy. The 13 single-amino-acids that define AAVv66 (relative to prototypical AAV2) are primarily located around the threefold axis of symmetry. This region is widely accepted as the principal determinant of receptor interaction differences that define serotype tropism and host immunity[2]. As such, the most successful efforts to engineer AAV capsids for unique tropism pattern and immune recognition hinge upon modifications to this region[2]. However, such efforts often result in variants that are deficient in vector packaging and production, including the top performing engineered capsids[44]. This limitation undercuts their translational value as therapeutic platforms. Interestingly, we found that AAVv66 is not strongly neutralized by pre-immunization of mice with AAV2, suggesting at the potential for AAVv66 in gene therapy strategies that require redosing of vector (i.e., following initial treatments with rAAV2). In contrast, sera from pre-immunization of mice with AAVv66 can block AAV2 transduction in vitro. This finding suggests that AAVv66 may have a better capacity to evade adaptive immunity than AAV2, or that the antibodies deployed against AAVv66 in mice used in this study were at epitopes with shared topological features. Further evaluation for AAVv66's capacity to elicit the host immune response is needed. Identification of the key AAVv66 residues

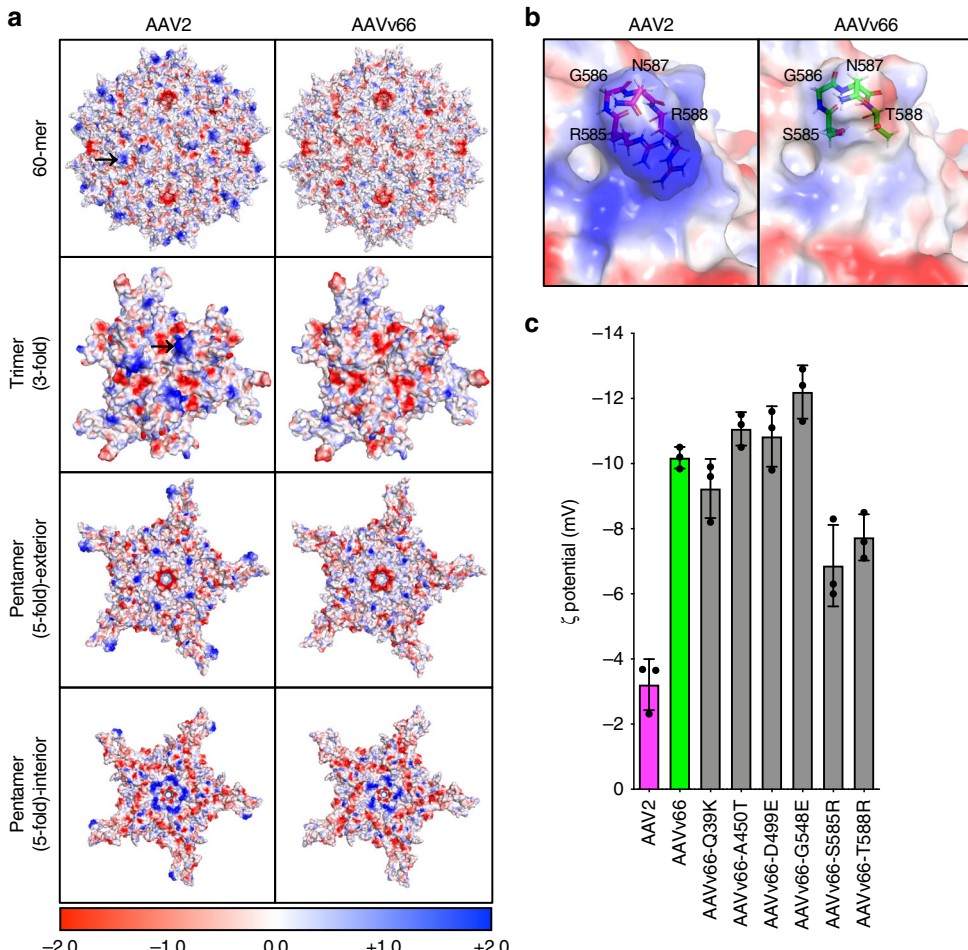

**Fig. 7 Differential capsid surface electrostatics between AAV2 and AAVv66. a** Surface positive (blue) and negative (red) charges are displayed for AAV2 and AAVv66 60-mer, trimer (threefold symmetry), and pentamers (exterior and interior of the fivefold symmetry) structures. Black arrows at the AAV2 60-mer and trimer structures indicate the approximate positions of R585 and R588 at a single threefold protrusion. The color scale bar represents the electrostatic potential on the solvent accessible surface in kT/e units. **b** Zoom-in of amino acid residues at 585–588 of AAV2 and AAVv66. **c** Bar graphs of the zeta potentials of purified vectors as measured by a Zetasizer. Values represent mean ± SD, $n = 3$.

identified in this study may aid in the engineering of AAV capsids with both altered tissue tropism and improved efficacy.

Perhaps the most remarkable difference from AAV2 is AAVv66's broad tissue spread following intrahippocampal injections, indicating its potential for CNS gene therapy. This difference could in part be due to reduced heparin binding. Reduced dependence on heparin, in turn, is likely advantageous for AAVv66 distribution, whereas high expression of HSPG on cell surfaces and the extracellular matrix impedes the spread of AAV2[45]. Efficiently targeting the hippocampus is critical for potentially treating epilepsy, Alzheimer's disease, or traumatic brain injury. Almost 10% of the population will experience a seizure in their lifetime and about 65 million people world-wide suffer from epilepsy[46]. Surgical removal of the epileptogenic temporal lobe and hippocampus are still the current treatment of choice[47,48]. The transduction pattern of AAVv66 might be ideal to efficiently target hippocampus-related pathologies by stereotactic administration and would thus allow for brain region-specific treatments. We also found that AAVv66 transduction of the brain is somewhat limited, following facial-vein injection of neonatal mice (Supplementary Fig. 7), but is superior to that of AAV2. This finding suggests that AAVv66 has some ability to bypass the BBB, but not as efficiently as AAV9. It was recently found that AAV9 and AAV.PHP.B, a highly neurotropic engineered capsid, traverses the BBB via Ly6A receptors[49]. A possible

explanation for our findings may be that the unique amino acids that define AAVv66 might bind and utilize a receptor besides Ly6A to weakly traverse the BBB. Further exploration into this possibility is indeed warranted.

The capsid stability of AAVv66 and its ability to confer genome packaging and release are significantly enhanced compared to prototypical AAV2. We provide evidence that the packaging yield of AAVv66 is only partially dependent on the stability of the capsid, since only one residue (D499) contributes to both stability and packaging. As expected, vector production likely depends on many factors, including receptor binding. Disruption of heparin binding may prohibit viral reinfection in vitro[24], thereby increasing the abundance of viral particles that can be purified from the supernatant during vector production (Supplementary Fig. 2). In addition, we conclude that AAVv66 differs from AAV2 in pH-dependent capsid functions, which are key in virus propagation[50]. Our findings suggest that vector genome release under acidic conditions may occur before complete capsid disassembly in a fashion that is similar to some other serotypes[35]. Notably, AAVv66 demonstrates higher propensity for genome release than AAV2 at acidic pHs. These observations suggest that while AAVv66 capsid is more thermally stable than AAV2 under acidic environments, like those found in late endosomes, DNA release by AAVv66 may be more efficient than AAV2. These findings may indicate that AAVv66 capsid uncoating is regulated

by specific compartmental conditions (e.g. within late endosomes or lysosomes), that are distinct from AAV2. The implications of these biophysical properties for gene therapy require further investigation.

Structural investigation via cryo-EM analysis and functional analysis by mutagenesis have provided insights into the structural stability of AAVv66. We found that the AAVv66-specific D499 and S585 amino acid residues are the most critical for capsid stability (Fig. 4d). D499 facilitates interactions at the interface between VP3 monomers (Fig. 5d). S585, together with T588 strongly affect the electrostatic properties of the capsid (Fig. 7c), suggesting that redistribution of the net charge on the capsid surface relative to that of AAV2 contributes to the overall structural stability. These findings correlate with alanine scanning mutation studies of AAV2, which revealed that disruption of HSPG-binding increases overall capsid stability[51].

Finally, we would like to note that at the time of our characterization of AAVv66, Tordo et al.[52] reported the engineering of a capsid called AAV2 true type (AAV-TT), which is a rationally designed capsid based on unique amino acid differences observed among AAV2 variants identified in human pediatric tissues[53]. Interestingly, AAVv66 and AAV-TT share nine amino acids in common, eight in the VP3 domain and one in the VP2 domain. AAVv66 and AAV-TT may therefore share common biological features. It was reported that AAV-TT has a better vector spread following intracranial injection than AAV2, similar to our findings in by intrahippocampal injections. Direct comparisons between AAVv66 and AAV-TT may reveal functional differences between the two capsids. Nevertheless, the naturally derived AAVv66 exhibits strikingly different biophysical performance and in vivo tropism relative to those of AAV2. AAVv66 may serve as a novel gene therapy vector, and our findings pave the way for engineering other improved viral vectors for gene therapy.

## Methods

**Clinical sample collection and NGS.** For DNA extraction from human tissues, approval was first obtained from West China Hospital Institution Ethics Committee, a pancreatic neoplasm sample was acquired from a 71-year-old female patient following tumorectomy and pathology of the tissue by frozen section examination and intraoperative frozen section diagnosis (Department of Oncology, West China Hospital of Sichuan University, Chengdu, China). The specimen was de-identified in abidance to the HIPAA Privacy Rule and coded such that the identity of the individual was not able to be readily ascertained. The sample was stored in liquid nitrogen until DNA extraction. To avoid AAV DNA cross-contamination, DNA extraction, and PCR procedures were performed in a sterile UV-irradiated biosafety cabinet. All surfaces and equipment were sprayed with DNA-Exitus Plus (Applichem, Cat No: A7089) and wiped clean with Milli-Q water after 15 minutes. Frozen tissues were then thawed at room temperature, quickly cut to about 25 mg of tissue with disposable scalpels and placed in a 2 mL tube. Extractions of DNA from tissues were performed using the QIAamp DNA Mini Kit (Qiagen, #51306) according to manufacturer's recommended procedures.

For long-read SMRT sequencing and bioinformatics, amplicon libraries were generated from genomic DNA by standard PCR procedures. To amplify AAV genomes, PCR was performed on Platinum™ PCR SuperMix High Fidelity (Invitrogen) with the following cycle conditions: 97 °C for 1 min, 46 cycles of 98 °C for 10 s, 60 °C for 15 s, and 68 °C for 2 min 30 s; and 68 °C for 10 min. Correctly sized PCR products were gel-purified with PureLink™ PCR Purification Kit (Thermo Fisher) and used for a second round of 15-cycle PCR for barcoding. The primer pairs used were:

First round primers:
CapF 5′-GACTGCATCTTTGAACAATAAATGA-3′
CapR 5′-GAAACGAATTAACCGGTTTATTGATTAA-3′
Second round primers:
EF 5′-CATCACTACGCTAGATGACTGCATCTTTGAACAATAAATGA-3′
ER 5′-TAGTATATCGAGACTCGAAACGAATTAACCGGTTTATTGATTAA-3′

Amplicons representing capsid variant ORFs were subjected to standard SMRT sequencing library generation by the University of Massachusetts Medical School Deep Sequencing Core (UMASS DeepSeq Core). Sequencing was performed on the RSII platform. SMRT sequencing returned 17,727 DNA reads that mapped to the AAV2 *cap* ORF using the BWA-MEM algorithm[54]. To rule out artifactual sequences, reads were then filtered to exclude those that were less than 1800 nt and

more than 2500 nt in length, and then filtered on the quality of reads (Phred score >30). This filtering reduced the reads to 16,681. Finally, reads were processed through InDelFixer (https://github.com/cbg-ethz/InDelFixer) to remove single nucleotide insertions and deletions that may result from error-prone PCR or sequencing error. In order to consider only unique capsid sequences and to rule out low-confidence variants, we performed de novo assembly (Geneious R9) on the filtered reads to cluster reads with 99% of sequence similarity. Only read clusters represented by at least ten reads were considered unique DNA capsid sequences. DNA sequences were then translated to amino acid sequences to define the final list of unique AAV capsids.

Full AAV *cap* ORFs from contemporary AAV serotypes (hu.2 used for AAV2/3) were obtained from NCBI and the predicted amino acid sequences were aligned using the MUSCLE algorithm[55], iterating until convergence was achieved. PhyML (version 3.3. 20190909) was then used to generate the phylogenetic tree using default parameters from within SeaView (version 4.7)[56] and then visualized via the Interactive Tree of Life online tool[57].

**Viral vector production and purification.** Viruses were produced using the triple-transfection method in HEK293 cells and purified by CsCl gradient centrifugation[30]. All vectors described were packaged with either the self-complementary AAV vector expressing enhanced green fluorescence protein (scAAV-*CB6-PI-EGFP*), single-strand vector expressing firefly luciferase (ssAAV-*CB6-PI-Fluc*), single-strand vector expressing secreted human alpha1-anti-trypsin (ssAAV-*CB7-CI-hA1AT*), or single-strand vector expressing LacZ. All transgenes are driven by the CMV early enhancer/chicken β actin (*CB6/7*) ubiquitous promoter[58]. The vector genome used to package AAV9 was scAAV-*CB6-EGFP*. In anecdotal evidence, this transgene cassette confers slightly lower expression than produced by the scAAV-*CB6-PI-EGFP vector* in neonatal mice and reduces transgene-related toxicity at the high vector doses used in our study.

**Luciferase assays.** Infection of HEK293 cells with AAV2, AAV3b, and AAVv66 vectors packaged with *CB6-Fluc* were performed in biological triplicates. Cells were cultured in Dulbecco's Modified Eagle Media (DMEM/High Glucose) (Hyclone, SH30022LS) with 10% fetal bovine serum (FBS, Hyclone) under standard cell culture conditions. Cells were lysed 48 h post-infection to assess the infectivity of vectors via luciferase activity using the Luciferase Assay System, (Promega, E1501) and measurements were performed on a Synergy HT microplate luminometer (BioTek, Winooski, VT). Values are expressed as relative light units.

**Heparin competition assay.** HEK293 cells were seeded onto 24-well plates (5E5 per well). After ~24 h, cells were treated with adenovirus at an MOI of 100:1 for 1 h. AAV2-*CB6-Fluc* or AAVv66-*CB6-Fluc* vector was incubated for 30 min at 37 °C in the presence of increasing quantities of heparin (Sigma-Aldrich, H3149-25KU) in 10% FBS in DMEM (Life Technologies, 11965126). The medium was removed from wells and replaced by the virus–heparin mix with serum free medium to a total volume of 500 μL for each well. Two hour after transduction, 500 μL of medium (20% FBS in DMEM) was supplemented into each well. Twenty-four hour post-transduction, cells were lysed and spun down. Totally, 10 μL of the supernatant was incubated with of 50 μL luciferase substrate and samples were analyzed immediately by plate reader (BioTek).

**Animal use.** Six- to eight-week-old male C57BL/6J mice (The Jackson Laboratory) were injected by IV, IM, or intracranial administration of test vectors. Mice were kept on a 12 h light/12 h darkness light cycle, at 70–74F with humidity at 35–46%. Mice were fed normal chow (ISO-pro 300 Irradiated Diets (#5P76). Mice subjected to IV injections were administered with vectors packaged with ssAAV-*CB6-Fluc* transgenes (1.0E11 vg per mouse), and mice were sacrificed 14 days post-injection. Mice subjected to IM injections of the TA muscle, were administered vectors packaged with ssAAV-*CB6-Fluc* transgenes (4.0E10 vg per mouse), and sacrificed 28-days post-injection. Every week, up until, and at time of sacrifice, animals were injected intraperitoneally with D-luciferin substrate (150 μg g⁻¹) (Gold Bio-technology, LUCNA) and sedated with isoflurane and luciferase activity was quantified using the IVIS SpectrumCT imaging platform (Perkin Elmer) with 1 min exposures. Image acquisition was performed using Living Image software (version 4.7). Tissues were harvested and luciferase activities were assessed by the Luciferase Assay System (Promega). Quantification of vector genome abundances were assessed by TaqMan qPCR on a ViiA 7 real-time PCR instrument (Thermo Fisher Scientific).

Mice subjected to intrahippocampal injections were administered with vectors packaged with scAAV-*CB6-EGFP* transgenes (3.6E9 vg per mouse). Unilateral injections were performed in the right hemisphere using a stereotaxic frame (Stoelting Co. Wood Dale, IL). Hamilton syringes (1207K95, Thomas Scientific) and needles (77602-06, Hamilton) were used for injection. The following relative coordinates were used for all intra-hippocampal injections: x: −1.5 mm, y: −2mm, z: −2mm. Neonatal mice (P0–P1) were subjected to facial injections with vectors packaged with scAAV-*CB6-EGFP* transgenes (4E11 vg per mouse). All animal procedures described in this study were approved by the UMass Medical School Animal Care and Use Committee.

**Immunostaining**. Four weeks post-injection, animals were transcardially perfused with 1× phosphate-buffered saline (PBS), followed by 4% paraformaldehyde (PFA). Brains were extracted and subsequently fixed in 4% PFA overnight at 4 °C. Brains were then immersed in 30% sucrose (prepared in 1× PBS), at 4 °C, until equilibrated in sucrose mixture. Brains were embedded in a 1:2 OCT (Tissue Tek, Torrance, CA) and 30% sucrose mixture, and cryo-sectioned at 40 μm (Cryostar NX70, ThermoScientific, Waltham, MA). Sections were permeabilized in 0.5% TritonX-100 for 1 h, blocked in 5% goat serum (10% normal goat serum, 50062Z, Life Technologies) for 1 h, and then incubated in primary antibody (anti-NEUN, 1:1000, EMD Millipore MAB377; anti-GFAP, 1:500, EMD Millipore MAB360; anti-OLIG2, 1:200, Abcam ab109186; anti-IBA1, 1:1000, Wako Chemicals NC9288364; anti-GFP, 1:800, Invitrogen A11122) overnight at 4 °C. Sections were washed three times in 1× PBS and incubated in secondary antibody (anti-mouse, Invitrogen A32744; or anti-rabbit, Invitrogen A32740) for 1 h at room temperature. Sections were washed three times in 1× PBS and mounted with Vectashield containing DAPI (Vector Laboratories, Burlingame, CA).

**Microscopy and analyses**. Brain section images were acquired on a Leica SP8 Lightning High Resolution Confocal (Leica Microsystems, Wetzlar, Germany). Global brain images (10× tiled brain sections) and high-magnification images (63× region specific areas) were collected at the same intensity and exposure thresholds for each respective magnification. For high-magnification images, 40–50 z-stack steps were collected at a 0.29 z-size. Analysis was performed using Imaris 9.3 Software (Bitplane Inc., Zurich, Switzerland). Each image was 3D rendered and thresholds were manually established. To ensure consistency, non-biased 3D rendering of total sub-anatomical EGFP volumes colocalized with DAPI volumes and cell type-specific stains were used as proxies for cellular counts and the number of positively transduced cells. Percent quantifications of the different cell types within each ipsilateral sub-anatomical region was conducted, followed by percentage quantification of each cell type. Percent transduction was determined by normalizing colocalized EGFP volume to total volume of cell type-specific staining within each region. Per cell-specific stain, n = 3 mice were analyzed. Statistical calculations for Fig. 2b was conducted in Prism 8 (GraphPad Software, Inc., San Diego, CA) and analysis was performed using Student's unpaired t test.

**DSF assays**. For capsid stability experiments, 5 μL SYPRO Orange 5000× (Thermo Fisher Scientific) was diluted in 495 μL PBS (Corning) to make a 50× stock. Totally, 45 μL of virus was mixed with 5 μL of 50× SYPRO Orange (final SYPRO Orange concentration was 5×). Fluorescence was quantified using a ViiA 7 real-time PCR instrument (Thermo Fisher Scientific) with the following parameters: samples were incubated at 25 °C for 2 min, followed by a temperature gradient (25–99 °C, 0.4 °C per step and held at each step for 2 min). To monitor the fluorescence of the SYPRO Orange at each temperature step, the ROX filter was used with no passive reference. To investigate the effect of pH on the melting temperature of AAV vectors, we adapted the protocol from Pacouret et al.[35]. Briefly, 5 μL of the virus vectors, 5 μL of 50× SYPRO Orange, and 40 μL of 0.6 M acetate buffer pH-adjusted from pH 7 to pH 4 were mixed. $T_m$ values reported in this study is defined as the max Δsignal/Δtemp detected between 25 and 95 °C. To investigate vector genome release, the SYPRO Orange dye was switched to SYBR Gold (Thermo Fisher Scientific).

**Site-directed mutagenesis**. To generate point mutations in the AAVv66 capsid ORF, the Q5 site-directed mutagenesis kit (New England Biolabs) and primer pairs for mutagenesis displayed in Table 1 were used.

**Cryo-EM**. For grid preparation, AAVv66 was prepared for cryo-EM on grids with a lacey carbon support film (01824G, Ted Pella, Inc.). First, the grids were washed with acetyl acetate and allowed to dry overnight. Next, the grids were glow discharged with 20 mA current with negative polarity for 60 s in a PELCO easi-Glow glow discharge unit. Totally, 3 μL of 1E13 vg mL$^{-1}$ AAVv66-CB6-Egfp vector in buffer (5% sorbitol, 0.001% pluronic acid F68 in PBS) was placed onto the grids loaded on a Vitrobot Mark IV (ThermoFisher) cryo-EM plunging apparatus. The grids were blotted for 6–6.5 seconds with Whatman #1 filter paper at 10 °C and 95% relative humidity prior to rapid freezing in liquid ethane.

For electron microscopy and image processing, a data set consisting of 2033 movies was collected using SerialEM (version 3.6, http://bio3d.colorado.edu/SerialEM/) on a Titan Krios electron microscope (FEI) operating at 300 kV and equipped with a Gatan Image Filter (GIF) and a K2 Summit direct electron detector (Gatan Inc.) using 0.5–2.2 μm underfocus. Totally, 50 frames per movie were collected, and 34 frames were used at 1.43e Å$^{-2}$ per frame for a total dose of 48.62e Å$^{-2}$ on the sample. Pixel size was 1.0588 Å on the sample. Movies were imported into cisTEM (version 1.0.0-beta) and were aligned with dose filtering and CTF parameters were determined[59]. Next, a total of 52,874 particles were automatically picked within cisTEM (characteristic and maximum radius: 130 and 140 Å). We note that both particles encapsulating vector transgenes and the small percentage of empty capsids were used to determine the final structure. Within cisTEM, an initial reference for alignment was generated from all particles using the *ab initio 3D reconstruction* function. This reference and all particles were iteratively refined using *auto refine* to obtain a 2.95-Å resolution map as determined from the FSC_part cutoff at 0.143. One round of per-particle CTF refinement in manual mode improved map resolution to 2.62 Å. Lastly, one round of beam tilt refinement and reconstruction improved map resolution to 2.46 Å. 3D classification did not improve the maps. The final map was B-factor sharpened by applying a B-factor of −32.92 Å$^2$ using the PHENIX (version 1.14-3260) *auto sharpen* function[60].

**Structure refinement**. Cryo-EM structure of AAV2 (PDB ID: 1LP3, https://doi.org/10.2210/pdb1lp3/pdb) was used as a starting model for structure refinement. Variant residues were modeled using PyMOL (The PyMOL Molecular Graphics System, Version 2.0 Schrödinger, LLC). The resulting AAVv66 model, containing 60 copies of VP3, was refined against the cryo-EM map using PHENIX[60]. Real-space simulated annealing and B-factor refinement in PHENIX resulted in a stereochemically optimal model[60]. The refinement results are summarized in Supplementary Table 2. The model was inspected, and figures were prepared using PyMOL.

**Measurement of zeta potential**. Vectors were diluted to a concentration of ~1.0E9 vg mL$^{-1}$ for zeta potential analysis using the Zetasizer Nano ZS system (Malvern). Totally, 500 μL of sample was added into a universal dip cell (Malvern). Before measurement, the system was stabilized for 2 min. Three measurements were recorded for each sample.

**Immunological studies**. 1.0E11 vg per mice of scAAV-CB6-PI-EGFP were intramuscularly administered into the left/right tibialis anterior of C57BL/6J mice. Four weeks later, 1.0E11 vg per mice of ssAAVv66-CB7-CI-hA1AT or ssAAV2-CB7-CI-hA1AT was delivered to the contralateral leg. Serum was collected at weeks 4–8 by facial–vein bleeds to assess NAb titers and A1AT levels by ELISA.

For NAb assays[61], Huh-7.5 (5.0E4 cells per well) were seeded onto a 96-well plate 24 h prior to transduction at 37 °C. Ad helper virus was then added at a multiplicity of infection (MOI) of 100:1 to the cell monolayer and incubated for at least an hour. Serial dilutions of serum and ssAAV2-LacZ or ssAAVv66-LacZ mixed solution were prepared in a V-bottom 96-well plate and incubated at 37 °C for 1 h. The serum-AAV mixed solution was then added to cells and incubated at 37 °C for 24 h. Cells were lysed and treated with beta-galactosidase substrate using the Galacto-Star One-Step Assay System (Invitrogen). Luminescence signal was detected by Synergy HT microplate reader (BioTek). The transduction inhibition

**Table 1 Primer pairs for site directed mutagenesis to generate point mutations in the AAVv66 capsid ORF.**

| Mutation in AAVv66 | Forward primer (5′-3′) (lowercase = mutated bases) | Reverse primer (5′-3′) |
|---|---|---|
| Q39K | AGAGCGGCATaagGACGACAGCA | GCGGGCTTTGGTGGTGGT |
| A151V | GCATTCTCCTgtgGAGCCAGACT | TCTACCGGCCTCTTTTTTCC |
| K447R | TTACTTGAGCagaACAAACGCTC | TACAGATACTGGTCGATC |
| A450T | CAAAACAAACactCCAAGCGGAAC | CTCAAGTAATACAGATACTGG |
| M457Q | AACCACCACGcagTCCAGGCTTC | CCGCTTGGAGCGTTTGTT |
| A492S | ATCAAAAACAtctGCGGATAACAACAACAGTG | ACTCGCTGCTGGCGGTAA |
| D499E | CAACAACAGTgaaTATTCGTGGAC | TTATCCGCAGCTGTTTTTG |
| Y533F | TGAAGAAAAAtttTTTCCTCAGAGCGGGGTTC | TCGTCCTTGTGGCTGGCC |
| D546G | TGGAAAACAAggcTCGGGAAAAA | AAGATGAGAACCCCGCTC |
| G548E | ACAAGACTCGgagAAAAACTAATGTG | TTTCCAAAGATGAGAACC |
| S585R | CAACCTCCAGagaGGCAACACAC | GTAGATACAGAACCATACTGCTC |
| T588R | GAGCGGCAACagaCAGGCAGCCA | TGGAGGTTGGTAGATACAGAACCATACTG |
| T593A | GGCAGCCACCgcaGATGTCAACA | TGTGTGTTGCCGCTCTGG |

test samples at each dilution was normalized by comparing to naïve mouse serum (Sigma-Aldrich, Cat. No. S-3509) at the same dilutions.

For A1AT ELISAs, a 96-well plate was first coated with anti-A1AT antibody (1:1000, A-0409 Sigma) at 4 °C overnight, and wells were incubated with blocking buffer (1% nonfat milk and 0.05% Tween-20 in PBS buffer) for 1 h at room temperature. 1:20, 1:200, and 1:2000 serum dilutions were performed using sample buffer (0.05% Tween-20 in PBS buffer) in a 96-well plate along with positive control (100, 50, 25, 12.5, 6.25, and 3.125 ng mL$^{-1}$ A1AT). After the plate was washed three times, sera were added into each well and incubated at 4 °C overnight. The plate was then washed three times and incubated with goat anti-trypsin-HRP antibody (1:5500 dilution in sample buffer) for 2 h. Before reacting with substrate, the plate was washed six times to remove all residual proteins. Lastly, ABTS substrates were added into wells and the signal was read by a Synergy HT microplate reader (BioTek).

**Reporting summary**. Further information on research design is available in the Nature Research Reporting Summary linked to this article.

## Data availability

The authors declare that the data supporting the findings of this study are available within the article and its Supplementary Information files, or are available from the authors upon request. Relevant source data are provided as Source Data file. Sequences for capsids and vector constructs described in this report can be made available upon request. The model and map of AAVv66 underlying Figs. 5–7 and Supplementary Fig. 12 is available at the RCSB Protein Data Bank PDB: 6U3Q and the Electron Microscopy Database: EMD-20630, respectively. The nucleic acid sequence for AAVv66 underlying Fig. 1c and Supplementary Fig. 1 has been deposited at NCBI GenBank under the accession code MT496778. Source data are provided with this paper.

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

## Acknowledgements
This work was supported by NIH (2R01NS076991, 5P01HL131471, 5P01HD080642, 5R01AI121135, and UG3HL147367), Department of Defense (W81XWH-17-1-0212), Gates Foundation (OPP1132169), Michelson Found Animals Foundation (226020), and partial support from Spark Therapeutics. We thank Maria Zapp, Ellen Kittler, and the UMMS Deep Sequencing Core; Chen Xu, Kang Kang Song, and the UMMS CryoEM Core; Roger Davis and Norm Kennedy for confocal microscopy help; and Gang Han for use of the Zetasizer Nano ZS system.

## Author contributions
H.L.H., A.B., and A.B.L. designed, executed, interpreted the experiments related to cryo-EM and drafted the paper; M.X., L.L., and G.X. designed, executed, and interpreted experiments relate to SMRT sequencing and initial characterization of proviral capsid sequences; A.L. and D.G. designed, executed, interpreted experiments, and wrote descriptions related to mouse brain sections and IF microscopy; J.L., A.L., and L.R. performed mouse injections and preparation of tissues; Q.S. produced all of the AAVs used in this work; Y.W. supervised the acquisition of human tissue samples; and P.W.L.T., A.K., and G.G. supervised the research and prepared the paper.

## Competing interests
G.G. is a scientific co-founder of Voyager Therapeutics and holds equity in the company. G.G. is an inventor on patents with potential royalties licensed to Voyager Therapeutics and other biopharmaceutical companies. G.G. and D.J.G. are scientific co-founders of Aspa Therapeutics Inc., and hold equity in the company. G.G. and D.J.G. are inventors on patents with potential royalties licensed to Aspa Therapeutics Inc., and other bio-pharmaceutical companies. Remaining authors declare no competing interests.
