## [Peer Review File · Nature Communications]

Reviewers' Comments:

Reviewer #1:

Remarks to the Author:

Remarks to author:

I have read this manuscript with great interest. Effective passage of blood-brain barrier (BBB) and establishment of infection in brain and neurological system are critical for the AAA therapeutic usage to treat neurological diseases. In this manuscript, Gao and colleagues identified a novel capsid AAVv66 from human brain sample by third-generation sequencing. Although AAVv66 shares a 98% sequence similarity with AAV2, it can much more effectively setup infection in brain in animal model. The study made a thorough comparison between novel AAVv66 and AAV2 in viral production, tissue tropism, serologic behavior and viral particle physical property with both in-vitro and in-vivo assays. And the analysis on structural information of AAVv66 further elucidate the molecular mechanism of major determinant amino acids contributing to the virus stability. The broad CNS distribution and better packaging performance of AAVv66 suggest its potential in CNS gene therapy. Also, the detailed structural analysis on AAVv66 can further leverage capsid engineering.

Concerns are outlined below:

First, in supplementary figure 5, I notice that the intravenous injection of AAVv66 seems to have a BBB penetrating property. Are there any data that can demonstrate the brain tissue AAV burden (e.g. qPCR) to support AAVv66 BBB penetration? If so, the differences between the BBB penetrating serotype AAV9 and AAV PHP.B / AAV PHP.eB are worthy to be addressed, or at least have a detailed discussion.

Second, could the transduction of AAVv66 and AAV2 be inhibited by pre-immunization with AAVv66? This evidence should be a necessary supportive data.

Third, I found it interesting that the amino acids which have most impact on viral particle stability or packaging efficiency are also amino acids interacting with various cellular receptor (e.g. E499 in AAV2 interacts with AAVR PKD2 domain; R585 and R588 in AAV2 interact with HSPG). However, these residues are not significantly involved in the inter-capsomer interaction to stabilize the virus particle. Could you raise a hypothesis that can possibly explain this phenomenon?

Forth (related to the third point), since a set of residues which distinguish AAVv66 and AAV2 are close to or overlapped with the residues that participate in the interaction with receptors (AAVR or HSPG), could it be possible that these residues affect receptor recognition, in addition to the impact on virus package? Moreover, a

very interesting thing is that the AAV variants derived from AAV9 pass BBB in an AAVR-independent manner, but is associated with Iy6a protein. I am very interested to know whether AAVv66 has AAVR-dependent passage of BBB, or not. And I am interested to know whether AAVv66 is AAVR-dependent to infect the normal cells, or not. Is there any difference with AAV9/PHP.eb and AAV2? Though I have some many questions on this point, I fully understand that this may require many experiments to be demonstrated. Since AAVv66 should rise great interests to AAV field with urgency, I suggest it could be possible that the authors give more discussion on this point, if they feel they could not provide these experiments in a short time.

Minor comments:

The punctuation marks should be used correctly. For example, the 3rd line of the last paragraph in Discussion "...Interestingly. AAVv66 and AAV-TT..." a comma should follow "interestingly" not a period.

The 3rd line of section "The AAVv66 capsid is more thermostable than AAV2 across a range of pHs", "uncoat" should be "uncoating".

Reviewer #2:

Remarks to the Author:

The manuscript by Gao and colleagues centers on the discovery and characterization of a natural AAV isolate from surgically excised human tissue. The new strain AAVv66 shows structural and functional properties that highlight three critical findings - (1) surgically excised human tissue is an invaluable resource for discovery of new AAV strains with clinical potential in gene therapy applications; (2) heparin binding properties and electrostatics affect viral spread within the CNS; and (3) provide a roadmap for structural manipulation of other AAV strains that might improve their functional properties. The new strain v66 should be of immediate interest to the CNS gene therapy community for application in epilepsy and other similar focal brain indications. The study is rigorous and technically robust - a comprehensive analysis of a new AAV strain ranging from discovery, biophysical and structural analysis as well as functional characterization in mice. Some subjective concerns are noted below. There are a number of new AAV strains (e.g., AAV-TT, AAV-retro, other AAV2 heparin binding mutants in general) currently under evaluation. It is somewhat well known in the virology field that lower affinity to viral receptors enables increased spread and virulence. Additional studies in non-human primates might be essential to determine whether these properties could translate into the clinic. Otherwise, a thorough and technically rigorous study of a new AAV isolate.

Response to reviewers

We appreciate the overall positive response from the reviewers and their recognition of our work's novelty. We also appreciate the reviewers' insightful comments and concerns, which have helped us improve the manuscript. In summary, we have expanded our investigation to include two additional mouse studies based on the questions that were raised. Below, we provide point-by-point responses to each comment. We have also marked within the revised the manuscript areas where changes have been made. During the process of revising the manuscript, we have also made revisions in sections of the manuscript that we felt required further explanation and clarification. These modifications and additions are also marked in red for review.

Reviewer #1 (Remarks to the Author):

Remarks to author:

I have read this manuscript with great interest. Effective passage of blood-brain barrier (BBB) and establishment of infection in brain and neurological system are critical for the AAA therapeutic usage to treat neurological diseases. In this manuscript, Gao and colleagues identified a novel capsid AAVv66 from human brain sample by third-generation sequencing. Although AAVv66 shares a 98% sequence similarity with AAV2, it can much more effectively setup infection in brain in animal model. The study made a thorough comparison between novel AAVv66 and AAV2 in viral production, tissue tropism, serologic behavior and viral particle physical property with both in-vitro and in-vivo assays. And the analysis on structural information of AAVv66 further elucidate the molecular mechanism of major determinant amino acids contributing to the virus stability. The broad CNS distribution and better packaging performance of AAVv66 suggest its potential in CNS gene therapy. Also, the detailed structural analysis on AAVv66 can further leverage capsid engineering.

Concerns are outlined below:

First, in supplementary figure 5, I notice that the intravenous injection of AAVv66 seems to have a BBB penetrating property. Are there any data that can demonstrate the brain tissue AAV burden (e.g. qPCR) to support AAVv66 BBB penetration? If so, the differences between the BBB penetrating serotype AAV9 and AAV PHP.B / AAV PHP.eB are worthy to be addressed, or at least have a detailed discussion.

Response: Thank you for this observation. In Supplemental Figure 5, these mice are actually lying ventral side up. The figure legends were not sufficiently complete, which may have led to the confusion. Because the mice were positioned with the ventral side up, we do not believe this is signal from the brain region. Rather, it is likely coming from the jaw/snout area. To address this concern, we have revised the legend for Supplementary Figure 5.

Furthermore, this reviewer's question (and the related one below) led us to directly test whether AAVv66 has the capacity to traverse the BBB (**Supplementary Figure 7**). We injected AAVv66 vector packaging the *EGFP* reporter via facial-vein administration and compared brain expression with those conferred by AAV2 and AAV9 vectors. We found that AAVv66 does have some ability to bypass the BBB, but the resulting transduction is not as strong as AAV9. As a result of these new data, we have added the following statement to our discussion:

*"We also found that AAVv66 transduction of the brain is somewhat limited, following facial-vein injection of neonatal mice (**Supplemental Figure 7**), but is superior to that of AAV2. This finding suggests that AAVv66 has some ability to bypass the BBB, but not as efficiently as AAV9. It was recently found that AAV9 and AAV.PHP.B, a highly neurotropic engineered capsid, traverses the BBB via Ly6A receptors⁵¹. A possible explanation for our findings may be that the unique amino acids that define AAVv66 might bind and utilize a receptor besides Ly6A to weakly traverse the BBB. Further exploration into this possibility is indeed warranted."*

Second, could the transduction of AAVv66 and AAV2 be inhibited by pre-immunization with AAVv66?

This evidence should be a necessary supportive data.

Response: We thank the reviewer for this insightful comment. We have now performed a second neutralizing antibody (NAb) study as per the reviewer's suggestion. Interestingly, NABs generated by mice treated with AAVv66 can effectively neutralize both AAV2 ($\text{NAb}_{50} = 1/1,280$) and AAVv66 ($\text{NAb}_{50} = 1/2,560$) *in vitro*. It is possible that the NABs generated with AAVv66 epitopes can recognize both AAV2 and AAVv66, while NABs against AAV2 are more specific towards AAV2. Because of these results, we have changed the interpretation of our data and state that, "...the serology of AAVv66 is not entirely distinct from AAV2."

Third, I found it interesting that the amino acids which have most impact on viral particle stability or packaging efficiency are also amino acids interacting with various cellular receptor (e.g. E499 in AAV2 interacts with AAVR PKD2 domain; R585 and R588 in AAV2 interact with HSPG). However, these residues are not significantly involved in the inter-capsomer interaction to stabilize the virus particle. Could you raise a hypothesis that can possibly explain this phenomenon?

Response: We apologize for not being clearer with our data interpretations, and thank you for giving us the opportunity to clarify these points. As mentioned by the reviewer, our capsid stability assay, revealed that the individual D499E and S585R mutations can impact both packaging and capsid stability ($\Delta T_m = 3.8$ and 5.9 °C) by DSF analysis (**Figure 4c and d**). Hence, we think that these mutations can impact both receptor recognition and capsid stability. We hypothesize that this phenomenon can be explained by charge changes at these positions that may impact inter-capsomer stability. D499E is located at the interface of two capsomeres where the extended side chain has the potential to interact with the side chain of T448 (**Fig. 6c**). S585R and T588R located at VR-VIII are spatially related (**Fig. 6F**). The positively charged residues R585 and R588 can potentially repel the loop thereby decreasing the stability of the capsid. It is clear that this does not prevent the capsomers

from a forming stable capsid, since these residues are native to AAV2, but these differences in thermostability and packaging yield point towards how AAVv66 is intrinsically better than AAV2.

Forth (related to the third point), since a set of residues which distinguish AAVv66 and AAV2 are close to or overlapped with the residues that participate in the interaction with receptors (AAVR or HSPG), could it be possible that these residues affect receptor recognition, in addition to the impact on virus package? Moreover, a very interesting thing is that the AAV variants derived from AAV9 pass BBB in an AAVR-independent manner, but is associated with ly6a protein. I am very interested to know whether AAVv66 has AAVR-dependent passage of BBB, or not. And I am interested to know whether AAVv66 is AAVR-dependent to infect the normal cells, or not. Is there any difference with AAV9/PHP.eb and AAV2? Though I have some many questions on this point, I fully understand that this may require many experiments to be demonstrated. Since AAVv66 should rise great interests to AAV field with urgency, I suggest it could be possible that the authors give more discussion on this point, if they feel they could not provide these experiments in a short time.

Response: We gratefully appreciate your questions and suggestions. Yes, we also think that these residues ultimately impact virus packaging as mentioned above. However, we have not yet explored receptor binding for AAVv66. We aim to investigate this in future studies. As mentioned above, we have added to our report evidence that AAVv66 can weakly bypass the BBB. This new finding is indeed intriguing, since we found that this capsid has the capacity to spread well upon intrahippocampal injections. Perhaps crossing the BBB is only a partial barrier for this capsid. Again, a possible explanation for this result is that AAVv66 may use a receptor besides Ly6A to traverse the BBB. Further exploration is indeed warranted to address this hypothesis.

Minor comments:

The punctuation marks should be used correctly. For example, the 3rd line of the last paragraph in

Discussion "...Interestingly. AAVv66 and AAV-TT..." a comma should follow "interestingly" not a period.

Response: We thank the reviewer for catching this and other typos. They have been corrected throughout.

The 3rd line of section "The AAVv66 capsid is more thermostable than AAV2 across a range of pHs", "uncoat" should be "uncoating".

Response: We thank the reviewer for offering this suggesting. However, the use of "uncoat" here is more grammatically correct than "uncoating" in this instance. We have added "must" in front of "uncoat" to avoid confusion.

Reviewer #2 (Remarks to the Author):

The manuscript by Gao and colleagues centers on the discovery and characterization of a natural AAV isolate from surgically excised human tissue. The new strain AAVv66 shows structural and functional properties that highlight three critical findings - (1) surgically excised human tissue is an invaluable resource for discovery of new AAV strains with clinical potential in gene therapy applications; (2) heparin binding properties and electrostatics affect viral spread within the CNS; and (3) provide a roadmap for structural manipulation of other AAV strains that might improve their functional properties. The new strain v66 should be of immediate interest to the CNS gene therapy community for application in epilepsy and other similar focal brain indications. The study is rigorous and technically robust - a comprehensive analysis of a new AAV strain ranging from discovery, biophysical and structural analysis as well as functional characterization in mice. Some subjective concerns are noted below. There are a number of new AAV strains (e.g., AAV-TT, AAV-retro, other AAV2 heparin binding mutants in general) currently under evaluation. It is somewhat well known in the virology field

that lower affinity to viral receptors enables increased spread and virulence. Additional studies in non-human primates might be essential to determine whether these properties could translate into the clinic. Otherwise, a thorough and technically rigorous study of a new AAV isolate.

Response: We thank the reviewer for the invaluable comments. We are currently pursuing direct comparisons of AAVv66 with AAV-TT, retro, and heparin mutants in both mice and NHPs. We note that AAVv66 is a natural variant and loss of heparin binding is only one hallmark of this variant. These previously characterized capsids have been key to how we have evaluated our variant as a promising clinical vector. We have added the following statement in our discussion to address the point raised by the reviewer:

“This difference could in part be due to reduced heparin binding. Reduced dependence on heparin, in turn, is likely advantageous for AAVv66 distribution, whereas high expression of HSPG on cell surfaces and the extracellular matrix impedes the spread of AAV2.”

Additionally, we agree that using non-human primates would further decipher the characteristics of AAVv66 as a gene therapy vector. These studies are currently underway and we look forward to reporting on this work.

Reviewers' Comments:

Reviewer #1:

Remarks to the Author:

The authors have responded most of concerns raised previously in this revision. Authors have demonstrated that AAVv66 has halved BBB penetrating ability compared to AAV9, while significantly stronger ability than AAV2. They also demonstrated that the serology of AAVv66 is partially distinct from yet similar with AAV2.

I still have some minor concerns:

- 1) The authors used neonatal mice(P0-P1) facial-vein injection method to compare AAVv66's and AAV2's BBB penetrating ability during systematic administration. But I am very curious that would there be any alternate outcomes when administrate these vectors with adult mice tail-vein injection? Considering there have been reports demonstrated that administrating AAV-9 at different development timepoint (neonatal or adult mice) could lead to different cell type transduction or tissue tropism. Some reference are appended here: (1) doi: 10.1038/nbt.1515 ; (2) doi:10.1038/sj.gt.3303029
- 2) Since authors propose that the side chain of 499E and 488T; 585S and 588T have potential interactions. I think it would be informative to provide molecular distances of these side chains in Fig.6c and Fig. 6f.

Comments:

The authors have responded most of concerns raised previously in this revision. Authors have demonstrated that AAVv66 has halved BBB penetrating ability compared to AAV9, while significantly stronger ability than AAV2. They also demonstrated that the serology of AAVv66 is partially distinct from yet similar with AAV2.

Response:

We thank the reviewer for accepting our revisions. We did our best to make significant improvements to the manuscript, following the suggestions.

Comment 1.

The authors used neonatal mice(P0-P1) facial-vein injection method to compare AAVv66's and AAV2's BBB penetrating ability during systematic administration. But I am very curious that would there be any alternate outcomes when administrate these vectors with adult mice tail-vein injection? Considering there have been reports demonstrated that administrating AAV-9 at different development timepoint (neonatal or adult mice) could lead to different cell type transduction or tissue tropism. Some reference are appended here: (1) doi: 10.1038/nbt.1515 ; (2) doi:10.1038/sj.gt.3303029

Response 1.

We thank the reviewer for raising this concept. We are (and were) very much aware of the different age-related transduction outcomes in mice. The reasons we chose to perform neonatal injections are manifold. Please allow us to describe the top two most relevant reasons here:

- a) We anticipated that the transduction of the CNS is weak when delivered into adults via tail vein injections, even for AAV9 (please compare results from doi: 10.1038/nbt.1515 with our own Figure S7g-h data. We aimed to clearly demonstrate to the readers the transduction efficiencies between AAVv66 and AAV9 following systemic delivery under best-case conditions without having to elaborate on the nuisances of administration differences for AAV vectors. In other words, we were very hesitant to show weak EGFP expression in the hippocampus, which is poorly targeted via tail-vein in adults, even by AAV9 (Figure 5m of doi: 10.1038/nbt.1515). To minimize any misperceptions coming from readers outside of the AAV field, we chose to perform neonatal injections to obtain data that correlated best with what we observed via intrahippocampal injections.
- b) We also chose to perform neonatal injections, since many CNS diseases, including Canavan disease, the main neurological disease we work on in the lab, are only treatable by AAV platforms at early infancy. This necessity is because neurological diseases are very damaging to the brain, and current gene therapy strategies can only halt or slow the disease, but cannot reverse it. In short, treatment at neonatal stages are more clinically relevant to CNS diseases than adult treatments.

Comment 2.

Since authors propose that the side chain of 499E and 488T; 585S and 588T have potential interactions. I think it would be informative to provide molecular distances of these side chains in Fig.6c and Fig. 6f.

Response 2.

We have updated Figure 6 panels (c and f) to display distances as thoughtfully suggested by the reviewer. Recognizing that these may be difficult to discern to the reader, we have also included Supplementary Table 3, which lists the specific distances highlighted in Figure panels 6c and 6f. I hope these will be informative enough for the readers.